# A multifunctional soft robotic shape display with high-speed actuation, sensing, and control

B. K. Johnson [1,6], M. Naris [1,6], V. Sundaram[1], A. Volchko[1], K. Ly[1], S. K. Mitchell[1,2], E. Acome[1,2], N. Kellaris[1,2,3], C. Keplinger [1,3,4] ✉, N. Correll [5] ✉, J. S. Humbert [1] ✉ & M. E. Rentschler [1] ✉

Shape displays which actively manipulate surface geometry are an expanding robotics domain with applications to haptics, manufacturing, aerodynamics, and more. However, existing displays often lack high-fidelity shape morphing, high-speed deformation, and embedded state sensing, limiting their potential uses. Here, we demonstrate a multifunctional soft shape display driven by a $10 \times 10$ array of scalable cellular units which combine high-speed electro-hydraulic soft actuation, magnetic-based sensing, and control circuitry. We report high-performance reversible shape morphing up to 50 Hz, sensing of surface deformations with 0.1 mm sensitivity and external forces with 50 mN sensitivity in each cell, which we demonstrate across a multitude of applications including user interaction, image display, sensing of object mass, and dynamic manipulation of solids and liquids. This work showcases the rich multifunctionality and high-performance capabilities that arise from tightly-integrating large numbers of electrohydraulic actuators, soft sensors, and controllers at a previously undemonstrated scale in soft robotics.

Shape displays, shape morphing surfaces, or shape-changing interfaces are a class of robotic devices which generate surface geometries through actuation (shape morphing)[1–19]. Applications include information displays[2–4], human interaction[2–9], and manipulation of objects[3,10,12,13] or aerodynamics;[14,20] often, a surface with both spatial and temporal shape morphing control is capable of multiple applications[1–3,7,8], suggesting rich potential for multifunctionality in this domain. Shape morphing can be induced through a variety of methods including push-pin actuator arrays which either form the surface directly[3,8] or manipulate an elastic layer[5,13], hinge-actuated surface elements[4,9], particle jamming[16], pneumatic cells[11,12], and magnetic-driven[7,17–19] or thermal-driven[6,7,9,10,15] morphing of in-surface elements. However, these existing approaches typically face multiple drawbacks which limit their proposed applications, including (i)

surface discontinuities[3,8,10] or high surface temperatures from actuation[6,7,10] which limit haptic interaction; (ii) low fidelity of possible surface geometries[4,5,9,10,13] which limits interaction with both objects and human users; (iii) the requirement of large external devices like magnetic plates, tracking systems, or pumps[3,8,12,17–19] which reduces device viability in certain environments; (iv) slow reversible shape morphing[5–13,16,19] which impacts surface refresh rates and object manipulation; and (v) no embedded in-surface means of state feedback[5,6,8,10–14,16–19] which leaves the surface unable to respond to external stimuli like pressures or deformations.

A promising approach to address these limitations is to integrate high-speed soft robotic actuators and sensors with natural mechanical compliance[21–25] to form a shape display with embedded sensing and control. The advantages of compliance, robustness, and embodied

[1]Paul M. Rady Mechanical Engineering, University of Colorado Boulder, Boulder, CO, USA. [2]Artimus Robotics, Boulder, CO, USA. [3]Materials Science and Engineering Program, University of Colorado Boulder, Boulder, CO, USA. [4]Robotic Materials Department, Max Planck Institute for Intelligent Systems, Stuttgart, Germany. [5]Department of Computer Science, University of Colorado Boulder, Boulder, CO, USA. [6]These authors contributed equally: B.K. Johnson, M. Naris. ✉e-mail: ck@is.mpg.de; nikolaus.correll@colorado.edu; sean.humbert@colorado.edu; mark.rentschler@colorado.edu

intelligence of soft robotic materials have already been demonstrated in adjacent fields like biomedical devices, wearable technology, and human-robot-interaction[22,23,25–31]. In particular, soft electrohydraulic actuators exhibiting high-speed, high-force deformation have already shown effectiveness in braille and haptic interfaces[31–33]. In addition, embedded state sensing of electrohydraulic actuators has been demonstrated through multiple modalities[34–36]. However, integration and control of electrohydraulic actuators and soft sensors in high-dimensional arrays still remains challenging. While some approaches have demonstrated actuation of electrohydraulic arrays[31,33,37,38] they lack embedded feedback sensors, in turn limiting their performance and capabilities. Integrating sensor arrays is challenging due to electromagnetic interference of high voltage (HV) driving signals and signal collection[31,34,35], and integrated closed-loop sensor feedback has so far not been demonstrated at the scales necessary for high-fidelity shape morphing[36,39,40].

In this paper we introduce a new form of multifunctional shape display composed of scalable cellular units which tightly integrate soft actuation, embedded deformation sensing, and control (Fig. 1). Each cell is driven by a Hydraulically Amplified Self-healing Electrostatic (HASEL) actuator, a class of soft electrohydraulic actuator that exhibits high actuation frequency and specific power[41–43]. An interference-free magnetic-based sensor generates deformation feedback, and an elastic surface skin forms the interface between cells and the environment to maintain low surface compliance. We report several improvements in electrostatic charge control, distributed sensing, and control algorithms to enable individually addressable control over each cell. By repeating 100 cells in a 10 × 10 array we form a multifunctional soft shape display, addressing the aforementioned limitations of existing shape morphing surfaces (Fig. 1a and Supplementary Movie 1). Electrohydraulic actuation enables both high-fidelity shape morphing and rapid motion (Fig. 1a). We report high performance metrics for both shape morphing and self-sensing of the soft display: a 200 Hz control rate, up to 50 Hz actuation speed, sensing of deformation with 0.1 mm resolution and of force with 50 mN resolution. By embedding sensors directly at the surface layer, we gain the ability to detect both surface deformation and external forces. This enables novel capabilities not previously reported in shape morphing structures such as a self-displaying scale and user-driven drawing without the use of camera systems (Fig. 1a). We also demonstrate other high-performance applications like the precise and rapid control of ball dynamics on the surface via new algorithms (Fig. 1a). Our approach addresses the

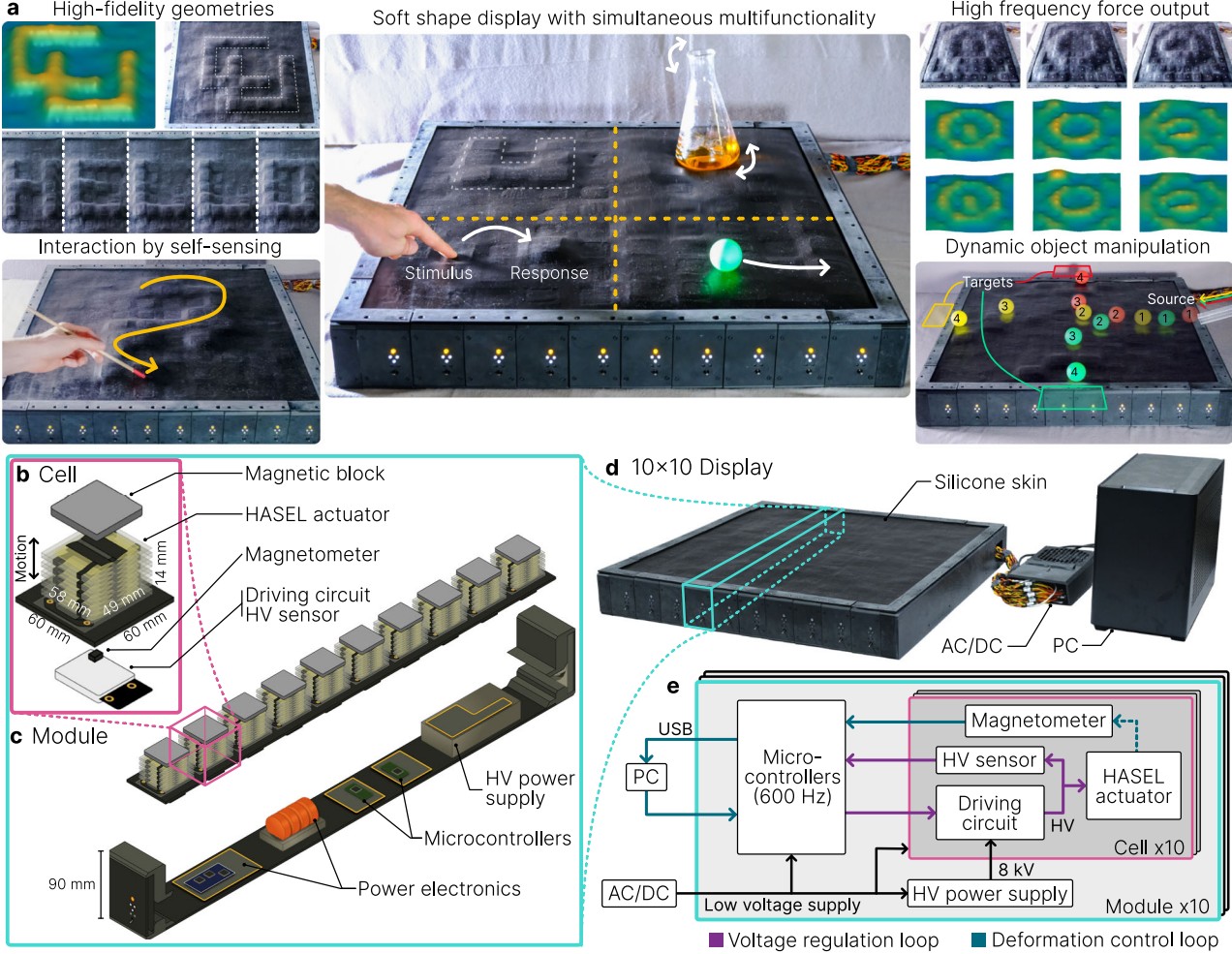

**Fig. 1 | Capabilities and design of the soft shape display. a** The soft shape display generates high-fidelity geometries (upper left), outputs motion and force at high frequencies (upper right), enables new interactions with embedded self-sensing (lower left), and dynamically manipulates objects in the environment (lower right). Multiple functions can be performed simultaneously across the surface (center). **b** The display is driven by a repeatable cell which combines a Hydraulically Amplified Self-healing Electrostatic (HASEL) actuator, magnetic-based sensing, and a controllable driving circuit. **c** 10 cells are arranged into a module which shares power and communication between cells. **d** 10 modules form the 10 × 10 array shape display, and a silicone skin forms the soft surface. A PC and AC/DC power supply respectively provide computation and power to the display. **e** Signal layout between components in the display. The cell components, module microcontrollers, and PC form control loops for HASEL voltage regulation and surface deformation which drive shape morphing.

key weaknesses of existing shape morphing surfaces and demonstrates a significant advancement in the scale of integration, sensing, and control of soft electrohydraulic arrays, leading to a multitude of high-performance capabilities for shape displays and other soft robotic devices.

## Results

### A scalable cellular unit with actuation, sensing, and control

To drive the soft shape display, we create a scalable cell (Fig. 1b) which tightly integrates actuation, sensing, and control; an approach similar to other large-scale robotic materials[44,45]. Each cell induces shape morphing via linearly expanding HASEL actuators which transduce electrical energy to mechanical deformation by electrohydraulic principles[43,46]. HASELs offer high actuation speeds, low steady state power use, and high specific power[42,43,46,47]. In comparison, shape displays driven by other soft actuators like shape memory alloys (SMA), liquid crystal elastomers (LCE), or pneumatics face limitations such as high heat generation[6,10], slow reversible actuation[6,10,12], or large compressor systems[12]. In addition to actuation, each cell incorporates an embedded sensor to enable direct surface feedback. The magnetic-based sensing mechanism[36,48] is both accurate at high frequencies and decoupled from the HASEL electric field, which contrasts with previous strain sensing[34] and capacitive self-sensing[35] approaches.

Each HASEL actuator is 58 mm × 49 mm × 14 mm undeformed and forms a 60 mm × 60 mm cell size in the shape display. The embedded sensor also fits within the footprint. It is possible to fabricate smaller HASELs for smaller shape displays or increased pixel density, and it is also possible to increase the number of pouches in each actuator for greater surface deformation[49]. In addition to the actuator and sensor, each cell contains a driving circuit with an integrated HV sensor that regulates actuator voltage to achieve surface deformation. The combined HASEL actuator, sensor, and driving circuit form a complete cell (Fig. 1b).

To scale these cells into a 10 × 10 shape display we use a hierarchical hardware architecture which reduces the quantity and complexity of components. Ten cells repeat linearly to form a single row 1 x 10 module which shares power and computation among cells (Fig. 1c), with an overall height of 90 mm (includes 14 mm actuator height). 1 x 10 modules repeat to create the resulting 10 × 10 display, and an elastomeric skin stretches across the cells to form the continuous surface (Fig. 1d). The skin, actuators, and embedded sensors have low mechanical compliance, resulting in a soft shape display; however, actuation can modulate stiffness when desired[50]. The display is driven by a central computer (PC) and power supply which communicate with each module for global surface morphing and feedback control (Fig. 1e). While the device is currently powered through an external wall outlet, each cell has an average 2.8 W power consumption at peak load and each module draws roughly 1.6 A at 24 V, making it possible to power the device by battery; the soft display is self-contained and portable compared to those which require multi-camera systems, projectors, or other external features to operate[3,8,12,17–19].

### High-speed surface deformation by charge-controlled actuation

Each HASEL actuator consists of a stack of 12 liquid-dielectric-filled pouches with electrodes on both sides which transduce electrical energy to strain (Fig. 2a). Placed beneath the elastomeric surface skin, this strain results in surface deformation. The actuators operate between 0 and 8 kV with a typical maximum sustained deformation of approximately 12 mm at 8 kV after 60 cycles (Fig. 2b). This represents an 86% strain of the actuator and a 13.3% strain in comparison to the 90 mm cell height, similar to other pixel-based displays[3,8]. Figure 2b also shows hysteresis in the voltage-to-deformation relationship, as well as a decrease in deformation over multiple cycles due to the retention of electrical charges within the actuator's composite

dielectric structure[46]. Both effects are mitigated with feedback control[34].

To enable simultaneous independent driving of multiple actuators using a shared HV power supply, we implement an optoelectronic half-bridge driving circuit[40,51,52] in each cell (Fig. 2c and Supplementary Fig. 1). Each half-bridge uses two optocouplers, and each optocoupler consists of a low voltage infrared light emitting diode (LED) and an HV photodiode. Light from the LED modulates the resistivity and therefore current through the photodiode. In the half-bridge circuit, the charging optocoupler pulls current from the shared HV rail to the positive HASEL electrode and the draining optocoupler sinks current to ground. Tuning the pulse-width modulation (PWM) duty cycle to the LEDs of each optocoupler allows for fine-grained control of the current and the resulting voltage across the actuator (Fig. 2d and Supplementary Fig. 2). An HV sensor in the form of a voltage divider enables closed-loop feedback of the actuator voltage.

To design an effective closed-loop voltage regulator we measure the open-loop dynamics of the actuator and circuit system. Figure 2d shows that the relationship between duty cycle and HASEL charge rate through the half-bridge circuit can be linearized for control analysis. The system dynamics of voltage regulation are primarily driven by the electrical dynamics $G_e(s)$ (Fig. 2e) which lumps the dynamics of the half-bridge, HASEL actuator, and HV sensor circuits (Fig. 2c). To characterize the open-loop electrical dynamics we use frequency domain analysis across all 100 cells in the surface (see Methods), resulting in a transfer function from duty cycle $w$ (%) to HASEL voltage $v$ (kV) of

$$G_e(s) = \frac{293.58}{s}. \tag{1}$$

Using a loop-shaping approach (see Methods), we design a controller to regulate HASEL voltage. A key aspect of the control strategy is that only one optocoupler in the half-bridge is active at a time, which simplifies the electrical dynamics. The control law is solved independently and simultaneously for each cell in the soft display at 1 kHz and results in a voltage regulation bandwidth above 200 Hz across all 100 cells (Fig. 2f). Disturbance rejection of the controller occurs at a higher-than-expected frequency, suggesting the presence of additional unmodeled nonlinearities.

Using motion-captured surface deformation data and frequency domain analysis (see Methods and Supplementary Fig. 3), we also characterize the open-loop deformation response of all 100 actuators in the 10 × 10 display moving simultaneously. In this case, the closed-loop voltage regulator is a fixed component for the open-loop HASEL dynamics. The resulting experimental data is used to estimate the open-loop transfer function of the HASEL dynamics from voltage $v$ (kV) to deformation $z$ (mm) as

$$G_h(s) = 0.014 \frac{(24\pi)^2}{s^2 + 12\pi s + (24\pi)^2} \tag{2}$$

where $\pi$ is the mathematical constant. The actuation dynamics of each cell show a natural frequency of 12 Hz and become damped from 20 to 50 Hz (Fig. 2f), similar to previous dynamic experiments for this actuator geometry[34]. Damping is primarily due to the inertia of the dielectric fluid which dominates the dynamics beyond 20 Hz for this geometry[47]. This result also shows that the HV driving circuit and voltage regulation is sufficiently fast for the given actuator dynamics. Actuation up to 50 Hz can also generate haptic feedback, as Meissner and Pacinian corpuscle receptors in the fingertip can detect inputs starting at 10 and 40 Hz respectively[53].

The combined result across 100 cells is a high-speed, soft shape morphing surface. Multiple cells can coordinate local deformations to

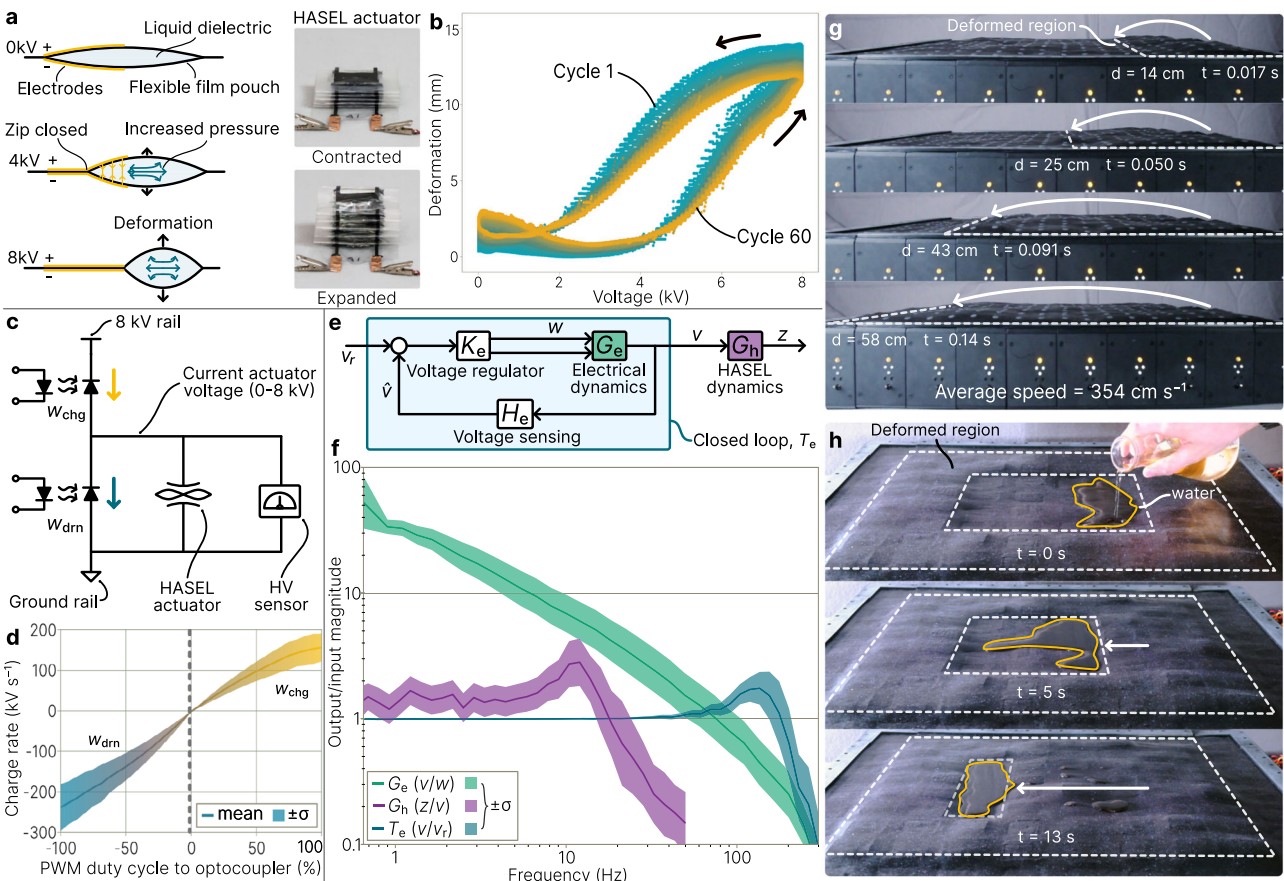

**Fig. 2 | Actuation characteristics of the shape display. a** (left) Basic dynamics of a single HASEL pouch. Application of high voltage (up to 8 kV) results in deformation. (right) A stack of pouches builds a folded HASEL actuator. **b** The voltage/displacement relationship for a typical HASEL over 60 cycles. **c** The half-bridge circuit which drives the actuator contains a charging and draining optocoupler, HV sensor, and HASEL actuator. **d** Relationship between input duty cycle ($w_{chg}$, $w_{drn}$ (%)) and actuator charge rate across 100 cells of the display. **e** Block diagram of the actuator voltage regulation loop with deformation $z$ (mm), voltage $v$ (kV), duty cycle $w$ (%),

voltage measure $\hat{v}$ (kV), and reference $v_r$ (kV). **f** Frequency response of the electrical dynamics, HASEL dynamics, and closed-loop blocks from (**e**) across 100 cells. The closed loop has a mean bandwidth of 200 Hz and the open-loop HASEL dynamics are below 50 Hz, providing sufficient control authority. **g** As a result of high frequency dynamics, the surface can generate traveling waves with speeds of 354 cm s$^{-1}$.
**h** Adjacent actuators can also drive the motion of objects on the surface, including liquid water.

generate global surface geometries like a horizontally-traveling wave, taking advantage of high actuation frequencies to generate waves with speeds up to 354 cm s$^{-1}$ (Fig. 2g). Because of the surface elasticity, the resulting wave is not rigidly pixelated despite being generated on only a 10 × 10 array. In addition to high-speed dynamics, the continuous and electrically-insulating skin allows the surface to safely manipulate liquids (Fig. 2h), a capability not demonstrated on other shape displays. Also, because the only moving components of the shape display are the actuators and silicone surface skin, high-speed shape morphing is very quiet; the combined source noise level of 100 cells deforming at resonant frequency is 8 dB above ambient, while slower operation is only 1–2 dB above ambient (Supplementary Movie 2). Furthermore, the system generates very little surface heat unlike thermal-based actuation, so the interface remains at ambient room temperatures – a useful property for human interaction or manipulating temperature-sensitive objects.

## Self-sensing of surface deformations and forces

In addition to high-speed actuation, each cell uses a soft magnetic block and magnetometer combination[36] to transduce actuation to measured deformation (Fig. 3a). Movement of the actuator or external deformation of the surface ($z$ (mm)) moves the magnetic block, and the magnetometer registers a change in magnetic flux density. Implemented across the surface, the result is a distributed magnetic

sensor array. A third-order polynomial mapping

$$\hat{z} = p_3 b_z^3 + p_2 b_z^2 + p_1 b_z + p_0 \qquad (3)$$

maps from the raw magnetometer sensor reading $b_z$ (mG) to the estimated cell deformation $\hat{z}$ (mm) using coefficients $p_0, \ldots, p_3$, which are unique to each cell. The magnetic data is sampled at 600 Hz and processed through a digital first-order low pass filter with a 50 Hz cutoff frequency. We calibrate the polynomial coefficients for each cell using motion capture data (see Methods).

Using this method, each cell in the soft display can measure deformation with a mean error below 0.1 mm for quasi-static deformation (Fig. 3b), resulting in a sensing resolution of 0.8% for 12 mm actuator deformation. Frequency response characterization also shows the sensors can accurately track HASEL deformations up to 30 Hz (Fig. 3c), while accuracy decreases above 30 Hz as the amplitude of deformation begins to approach the sensing resolution. The magnetometer measurements in each cell are not influenced by the magnetic flux of adjacent cells; however, the physical 10 × 10 surface boundary increases measurement inaccuracies along edge cells. Additionally, because the magnetometers in each 1 × 10 module are connected in series (see Methods), electrical impedance increases sensor noise for cells at the end of the module.

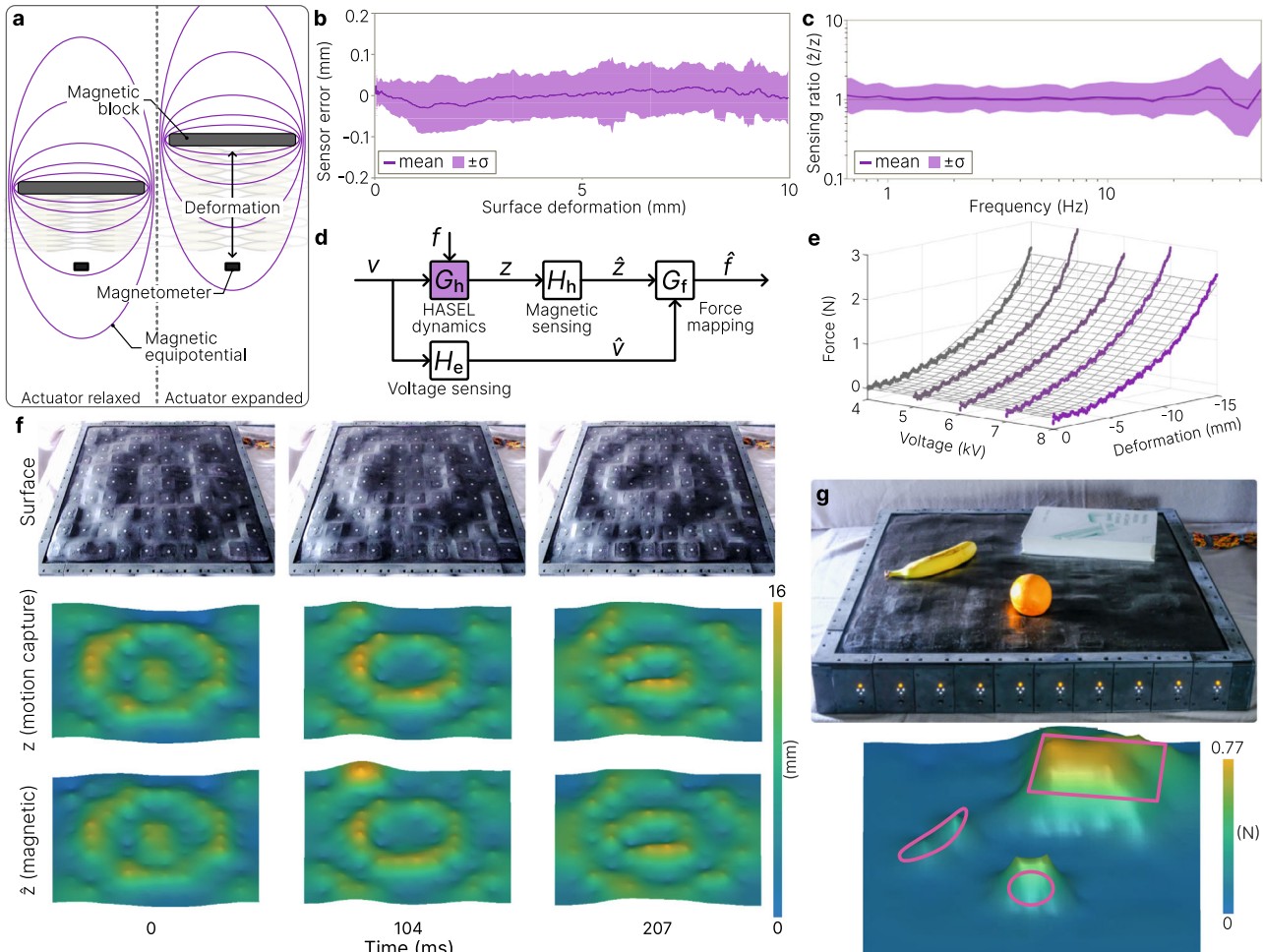

**Fig. 3 | Embedded self-sensing of deformation and force. a** Sensor functional mechanism. Deformation of the soft magnetic block causes a change in magnetic flux density measured by the magnetometer. A third-order polynomial maps from the magnetic flux to deformation. **b** Mapping error between the measured and true surface deformation of each cell ($n = 90$). **c** Frequency response ratio between the measured and true deformation of 100 cells. **d** Block diagram for cell force with actuator voltage $v$ (kV), measured voltage $\hat{v}$ (kV), deformation $z$ (mm), measured deformation $\hat{z}$ (mm), applied force $f$ (N), and estimated force $\hat{f}$ (N). **e** Experimental force-displacement curves at 4, 5, 6, 7, and 8 kV (solid lines) which are used to create the force mapping $G_f$ (meshed surface). **f** Time history comparing photographs with the true and measured surface deformations for 2 Hz shape morphing. **g** Objects placed on the surface are detected and mapped to estimated forces using the mapping in (**e**). The plotted surfaces in (**f**) and (**g**) are formed by quadratic interpolation between sensor values.

In addition to deformation sensing, the magnetometer also enables external force measurement when combined with HV sensor measurements from the HASEL driving circuit (Fig. 3d). Using a dynamic mechanical analyzer (see Methods), we quasi-statically load a set of actuators at set voltages and measured force and displacement data (Fig. 3e). Each actuator produces approximately 2.5 N for sub-mm deformations. The experimental data is fit to a 15-degree polynomial (Supplementary Eq. (1)) which maps the measured voltage and deformation to an estimated force. The resolution of the force mapping is within 50 mN, corresponding to a 5 g mass added to the surface (Supplementary Fig. 4).

When combined across all 100 cells of the soft display, the result is a magnetic sensor array capable of accurate deformation mapping at rapid time scales (Fig. 3f and Supplementary Movie 3). The soft shape display is able to self-sense surface deformation with a spatial resolution limited only by the number of cells. The display can also self-sense distributed forces, for example to detect the size and shapes of various objects on the surface (Fig. 3g). Due to the continuous elastic skin, forces applied across two adjacent cells are averaged across the cells. One can thus estimate the total force of an object by summing the distributed forces measured by each cell. Because the sensing mechanism is magnetic-based, deformation and force are detectable from nonconductive objects, in contrast to many haptic and touch displays which detect touch based on capacitive coupling between the device and object. Another advantage of the magnetic-based sensing over camera-based approaches is that it functions equally well in dimly-lit or object-occluded environments. The downside of this method is that the influence of external magnetic materials disrupts the magnetometer flux measurement, leading to inaccurate deformation mapping, and if the display is moved to a different location the sensors must be recalibrated.

## Surface shape morphing through feedback and external stimulus

Using the combined actuator and sensor arrays formed from 100 cells, we implement closed-loop feedback on the deformation of the surface. We apply the same loop shaping approach used for the voltage regulation controller (see Methods) to design an outer loop controller (Fig. 4a) for the desired deformation $z_r$ (mm) using the actuator dynamics in Eq. (2) The feedback loop runs at 200 Hz and is solved independently and simultaneously for each cell in the shape display, enabling the display to achieve accurate and fast shape morphing. The closed-loop deformation bandwidth is 20 Hz due to HASEL dynamics and signal latency; higher frequency actuation is achieved at reduced

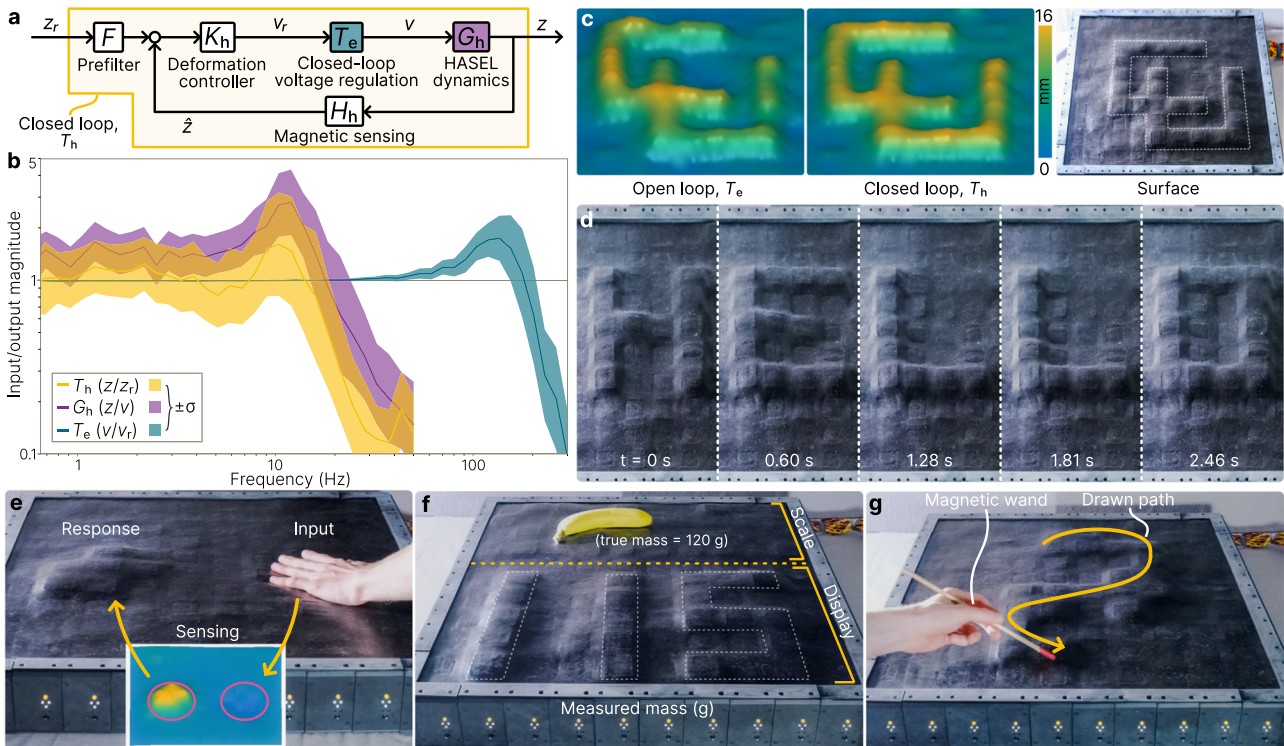

**Fig. 4 | Closed-loop feedback and interactivity. a** Block diagram of the closed-loop feedback on each cell with reference deformation $z_r$ (mm), deformation $z$ (mm), measured deformation $\hat{z}$ (mm), reference voltage $v_r$ (kV), and voltage $v$ (kV). The closed-loop voltage regulation $T_e$ is the block diagram shown in Fig. 3e. **b** Frequency response of the HASEL dynamics, closed-loop voltage regulation, and deformation-closed loop across 100 cells. The bandwidth of the deformation controller is limited to 20 Hz due to HASEL dynamics and communication latency. **c** The deformation controller results in greater precision of surface morphing compared to only voltage regulation (open loop). Here, the University of Colorado CU logo is plotted on the display surface. **d** Demonstration of a high-speed scrolling text display using the controller. **e** Embedded self-sensing enables feedback from external stimuli. Here, deformation on one side of the array is converted to actuation on the other side. **f** Combining (**d**) and (**e**) results in an intelligent scale which can both sense forces and display the resulting estimated mass. **g** The magnetometers also enable the use of peripheral devices, like a magnetic-tip wand which can draw shapes through surface sensing and deformation. The plotted surfaces in (**c**) and (**e**) are formed by quadratic interpolation between sensor values.

amplitudes (Fig. 4b). The controller not only rejects internal disturbances like charge retention on the HASELs but also external disturbances applied to the surface. As a result, the controller provides more accurate control of surface geometry when compared to actuator voltage regulation alone (Fig. 4c). Combining the closed-loop accuracy with high-speed HASEL actuation enables rapid morphing of time-varying surfaces like scrolling text (Fig. 4d and Supplementary Movie 4).

In addition to more accurate shape morphing, the combined actuating, sensing, and control capabilities of the soft robotic display enable new functions not demonstrated on similar devices. The surface can react to external stimulus like human touch, for example by mirroring the input as a proportional actuator response (Fig. 4e and Supplementary Movie 5). This stimulus-to-response capability has applications in haptics, object interaction, and teleoperated human interaction[3]. We can also further this principle by using the surface in parallel as a text display: for example, in an interactive scale which displays an object's mass in real time (Fig. 4f and Supplementary Movie 6). One region of the surface acts as the scale using embedded force sensing, and the other region acts as a text display which displays the mass (in grams) of detected objects based on the summation of forces across the scale region. This simple implementation of a novel functionality also highlights the display's ability to perform simultaneous local operations; functionality is not limited to global surface morphing with a single output (Fig. 1a and Supplementary Movie 7).

The embedded sensors also enable the use of peripheral devices that interact with magnetic fields. While similar in nature to existing magnetic-based surfaces[7,18], our approach offers high-fidelity shape control in real time via user-driven motions. For example, the movement of a magnetic-tip wand across the sensor array can be detected and converted into active surface morphing. This allows a user to interactively draw with the soft display by creating local changes in surface geometry (Fig. 4g and Supplementary Movie 8). In this approach, the display acts as a distributed magnetic flux sensor which can simultaneously form a physical representation of the flux density, creating an intuitive visualization of unseen forces. By interpolating between sensor values, the position of the wand tip can also be determined at a resolution greater than the cell resolution, suggestion potential use of the surface as an input tool for digital drawing or computer interfaces.

## Dynamic object conveyance

Like other shape morphing surfaces, the soft shape display is also capable of object manipulation on its surface[3,8,10,12,13,54,55]. To manipulate a round object like a ball, we use shape morphing to transform vertical surface deformation into lateral rolling, an interaction with applications to manufacturing, object conveying, and sorting[3,10,12]. To demonstrate manipulation of a ball on the display surface we implement a novel control algorithm (see Methods) using position and velocity feedback. The basis of this control is the local deformation of the surface at the ball's position to drive rolling motion (Fig. 5a). Using a semicircle shape with a Gaussian side profile (Fig. 5b), the display forms a concave surface along which the ball rolls. The direction of roll is based on the local surface gradient.

We demonstrate precise closed-loop positional control of a ball via active shape morphing first by moving a table tennis ball through a

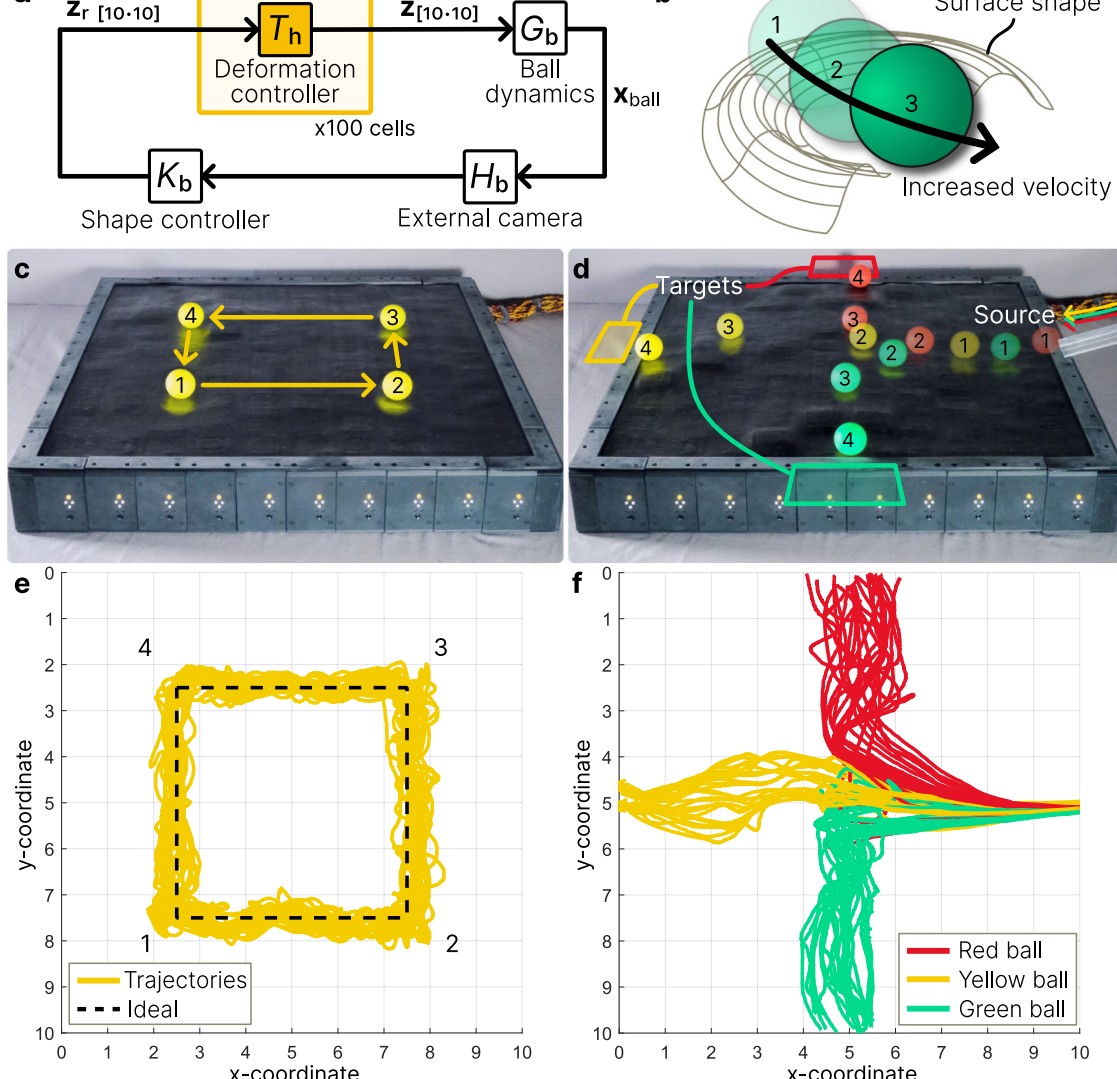

**Fig. 5 | Feedback-controlled object manipulation. a** Closed-loop control is achieved by integrating an overhead camera for ball detection and an algorithm to generate a driving surface shape (see Methods). The controller generates a $10 \times 10$ matrix of deformations $\mathbf{z}_{r[10 \cdot 10]}$ (mm) which corresponds to the deformation of each cell in closed-loop control. The achieved deformation $\mathbf{z}_{[10 \cdot 10]}$ (mm) results in a change to ball surface coordinates $\mathbf{x}_{ball}$ $(x, y)$. **b** Desired surface shape to induce ball rolling motion. Through the denoted sequence 1–3 the ball transforms potential to kinetic energy. **c** Planned motion sequence for a single ball on the surface. **d** Planned motion sequence to sort three balls by color. **e** Trajectories of 25 trials for the single ball. Each trial reaches each waypoint in an ideal straight line. The mean travel time to complete the square is 28.48 s with a mean speed of 1.424 cells s⁻¹ (85.44 mm s⁻¹). **f** Trajectories of 25 trials for sorting three balls by color. The mean sorting time from the source to the target is 4.66 s for each ball.

series of four waypoints, resulting in a square trajectory (Fig. 5c), and then by rolling three balls onto the surface and simultaneously sorting them into separate regions based on ball color (Fig. 5d). A monocular camera above the surface measures the ball position and detects multiple balls by color via image processing. Each experiment is repeated 25 times, and ball position data are collected for each trial; see Methods for full experimental setup.

For the square trajectory sequence, the ball trajectory over 25 trials is within 1/2 cell diameter of the desired trajectory (Fig. 5e), an error of 3 cm. The overall spread of trajectories has a width of about 1 cell, achieving the upper performance limit before the system is underactuated; we anticipate that increasing cell density of the display will lead to a smaller spread. The mean completion time was 28.48 s, resulting in a mean rolling speed of 1.424 cells s⁻¹ (85.44 mm s⁻¹), and the set of trajectories all maintain the desired square shape (Fig. 5e and Supplementary Movie 9). For the color sorting experiment, the

different-colored balls are all correctly sorted to their goal positions for all trials (Fig. 5f and Supplementary Movie 10) with a mean sorting time of 4.66 s for each ball. Because each ball during the sorting trials has an initial velocity on the surface, its trajectory is more susceptible to surface irregularities, resulting in a larger spread of trajectories compared to the single-ball experiment.

The results show enhanced performance over existing shape display demonstrations in object manipulation. Due to the high speed of reversible shape morphing, the display is able to drive ball motion dynamically by reacting to disturbances in the ball's position according to our control strategy. This contrasts with existing demonstrations of ball motion which typically use either quasi-station motion or trajectories enclosed within a tight channel to guide the ball[3]. Since this algorithm only generates local deformations which move with the ball, it is energy efficient in surface morphing; only about 10% of the surface is active. In addition, we show that the continuous soft surface enables

high controllability despite the ball diameter (40 mm) being smaller than the cell diameter (60 mm).

## Discussion

In this paper, we report multifunctional capabilities of a high-speed soft robotic shape display which combines soft electrohydraulic actuation, embedded sensing, and control. To achieve these results, we create a scalable cell with tightly-integrated actuation and sensing to form the 10 × 10 soft shape display and implement feedback control at a new scale for the soft robotics field, with a combined 100 independently-addressable electrohydraulic actuators, 100 soft sensors, and over 200 control loops for voltage regulation, deformation feedback, and global surface shape control. The integration of soft robotic components at this scale results in a multitude of emergent capabilities like high-fidelity geometry generation, traveling waves which can move both solids and liquids across the surface, complex user interactions, and more (Supplementary Movie 1). These demonstrations provide a flavor of the rich multifunctionality created by soft, self-sensing, shape morphing surfaces.

Our choice of electrohydraulic actuation provides significant performance in both speed and force output, while simultaneously being very quiet (Supplementary Movie 3) unlike pneumatic systems and without generating high temperatures like SMA- or LCE-driven surfaces. The actuators' mechanical softness also keeps the surface compliant for haptic interaction and object manipulation (Fig. 4) in contrast to rigid shape displays[3,4,8]. In addition to their high resolution, the embedded soft sensors also enable new functional modalities not possible in previous shape morphing surfaces. Measuring the deformation and force state across the surface allows for active response to external stimuli, such as human touch or magnetic stimuli from peripheral devices (Supplementary Movie 1 and Fig. 4). However, the sensors are susceptible to influences from external magnetic materials, limiting the types of objects that can be manipulated on the display. Because each cell is individually addressable, we also demonstrate spatial control of functionality (Supplementary Movie 7) – some cells can act as passive sensing units, while others can actively shape morph to display information to the user. Sensing capabilities can be expanded further by incorporating additional magnetometer axes to measure surface shear forces[56,57].

Since the display is built from a series of repeated cells, both the design and integration of each cell is important to enable scalability up to the 10 × 10 display and beyond. By sharing power and computation among cells in each 1 × 10 module, the hardware complexity and cost is reduced and computation load is offloaded from the central PC to local microcontrollers. While we repeat the single cell up to a 1 × 10 module, hardware limitations like circuit pin addressing, sensor signal impedance, and power consumption mean that the module cannot scale indefinitely. However, it is possible to create larger shape displays by increasing the number of modules. The biggest challenge of integrating many modules together is maintaining sufficient communication and computation speeds; new communication architectures will be required for larger-scale systems. Miniaturization of the cells is also potentially possible, but current techniques (see Methods) make the fabrication of smaller actuators (less than 25 mm × 25 mm) challenging. Performance at small cell sizes is also impacted due to actuator film thickness[43] and the loss of sensor resolution as each magnetic block emits a weaker field.

As the shape display represents a 100-actuator, 100-sensor soft robot, the hardware could also be applied to the design of other high-degree-of-freedom soft robots like continuum manipulators[58] or bioinspired systems[29]. The shape display we introduce in this paper thus demonstrates a wide variety of uses applicable to a multitude of scientific and industrial fields, and further shows the promise of using soft robotic materials for high-degree of freedom, high-speed, and sensor-rich robotic systems.

## Methods

### Hardware fabrication and software

The folded HASEL actuator and magnetic block in each cell are fabricated as described in the Supplementary Methods. The magnetometers (LIS3MDL, STMicroelectronics) are wired in series for each 1 × 10 module (Supplementary Fig. 5). Supplementary Fig. 1 shows the HV driver board. Photodiodes (OZ100SG, Voltage Multipliers Inc.) are soldered to the board and an insulating epoxy (Pratley White Epoxy, Pratley) is cast over HV components.

Each module is constructed from a set of 3D-printed plastic (XT-CF20 PETG, Colorfabb) and laser cut acetal components. The HV power supply (UltraVolt 10A24-P30, Advanced Energy) and circuit boards are attached to the acetal structure, and each HASEL and sensor placed on top of the structure to form the 1 × 10 array. Supplementary Figs. 6, 7, 8 show the module-level control and power boards. The ten modules in the display connect to the external power supply (ION SFX 650 G, Fractal Design) powered from a 120 V wall outlet. USB hubs collect the USB connections of all microcontrollers into two combined USB inputs to the PC. The silicone (EcoFlex 00-30, Smooth On) surface skin is cast in a single 550 μm layer.

The primary code to control the display was programmed in Julia[59] on the PC and C + + on the microcontrollers (Supplementary Software 1). Graphical plots are rendered with Makie.jl[60] and MATLAB R2021b (MathWorks). Further detail on hardware fabrication and software setup is described in the Supplementary Methods.

### Motion capture system

Supplementary Fig. 3 shows the 7-camera motion capture system (OptiTrack Prime 13 W) used to collect ground truth deformation data at 240 Hz. Circular retroreflective markers (Scotchlite 7610, 3 M) are placed in the center of each cell. The standard camera calibration is performed in Motive (OptiTrack) using a wand (CS-W500, OptiTrack) and calibration square (CS-200, OptiTrack), with a mean ray error below 0.5 mm and mean wand error below 0.2 mm. Positive z-coordinates correspond to upward surface deformation. Four markers in each corner of the 10 × 10 display are used to recalibrate the camera ground plane, setting the average z-coordinate of the marker positions to zero.

### Magnetometer deformation and force sensing calibration

To generate the deformation polynomial fit, each HASEL in the display is sent an input voltage ramp from 0 kV to 8 kV over 8 s. We sample the magnetometer reading and z-coordinate of the corresponding cell using motion capture data. The data are fed into a least-squares polynomial fitting function to generate the third-order polynomial coefficients which are unique to each cell. Calibration takes approximately 10 min. Because the sensor mapping is performed one cell at a time, inaccuracies are introduced when adjacent cells are actuated due to surface skin mechanical coupling. To account for this, the estimated deformation for a given cell $\hat{z}_{i,j}$ is modified to

$$\hat{z}_{i,j} = \left(p_3 B_z^3 + p_2 B_z^2 + p_1 B_z + p_0\right) + \alpha\left(\hat{z}_{i-1,j} + \hat{z}_{i,j-1} + \hat{z}_{i+1,j} + \hat{z}_{i,j+1}\right) \quad (4)$$

with scaling factor $\alpha = 0.05$. For edge cases (e.g., where $\hat{z}_{i-1,j}$ does not point to a valid cell), the adjacent $\hat{z}$ values are set to zero.

To create the force sensing map (Fig. 3e), a dynamic mechanical analyzer (DMA) (800E2, TestResources) applies incremental deformation to a HASEL actuator at a rate of 0.5 mm s⁻¹ to represent a quasi-static force. Prior to adding the force, the actuator receives a constant voltage of 4 kV and the baseline deformation is measured. The change in deformation and force are recorded from the DMA until a maximum of 25 N. This test is applied for HASEL input voltages of 4, 5, 6, 7, and

8 kV and using three different actuators. The mean data from all test actuators at each voltage is used to determine the polynomial surface fit in MATLAB R2021b Curve Fitting, resulting in a 15th order polynomial (Supplementary equation(1), Supplementary Table 1). The force map is the same for all cells because it assumes accurate deformation mapping.

## Frequency analysis and controller design

For the voltage regulation loop (Fig. 2e) we identify the relationship between the ADC signal of the HV sensor ($v_{raw}$) and the HASEL actuator voltage $v$ (kV) as

$$v = v_{raw}/102.82 \tag{5}$$

by measuring the HV sensor value for a set of HASEL voltages $\{0, 0.5, 1, \ldots, 7.5, 8\,\text{kV}\}$. To measure open-loop response (Fig. 2e), we input PWM signals

$$w(t) = 0.1\sin(2\pi f t) \tag{6}$$

with 32 logarithmically-spaced frequencies $f$ from 0.6 to 300 Hz and time $t$ from 0 to 15 s. The combined input $w$ maps to the duty cycle of each charging and draining optocoupler ($w_{chg}, w_{drn}$) via:

$$
\begin{aligned}
w_{chg}(t) &= \begin{cases} w(t), & w(t) \geq 0 \\ 0, & \text{otherwise} \end{cases} \\
w_{drn}(t) &= \begin{cases} |w(t)|, & w(t) < 0 \\ 0, & \text{otherwise} \end{cases}
\end{aligned} \tag{7}
$$

where $|\bullet|$ indicates absolute value. We assume that the HV sensor has unity gain ($H_e = 1$ and $v = \hat{v}$). The open-loop transfer function (Eq. 1) is fit to the data to match the gain at 200 Hz, which is the desired disturbance rejection frequency. Using the open-loop transfer function in Eq.(1) we design a controller

$$K_e(s) = \frac{3 \times 10^5}{s + 150} \tag{8}$$

such that $G_e K_e > 10$ for $f < 10$ Hz, $G_e K_e > 5$ for $f < 20$ Hz, and $G_e K_e < 0.1$ for $f > 200$ Hz. $K_e$ maps $v_{raw}$ to raw 16-bit values ($2^{16}$). The controller is implemented in a 1000 Hz loop using a zero-order hold (ZOH) discretization, with inputs sent at 600 Hz. The closed-loop bode plot (Fig. 2f) is generated with inputs $v_r(t)$ (kV) as

$$v_r(t) = 0.4(\sin(2\pi f t) + 1) + 0.8 \tag{9}$$

For the deformation feedback loop (Fig. 4a), the open-loop is characterized by inputs

$$v_r(t) = 0.4(\sin(2\pi f t) + 1) + 4 \tag{10}$$

with 32 logarithmically-spaced $f$ from 0.6 to 50 Hz and $t$ from 0 to 15 s. We assume $v = v_r$. Using the open-loop transfer function in Eq.(2) we design a controller

$$K_h(s) = 30\frac{(s + 70)}{s} \tag{11}$$

such that $G_h K_h > 20$ for $f < 10$ Hz, $G_h K_h > 5$ for $f < 20$ Hz, and $G_h K_h < 0.1$ for $f > 300$ Hz. The controller is implemented at 200 Hz using a ZOH discretization. We also add a prefilter

$$F(s) = \left(\frac{200}{80}\right)^2 \frac{(s + 80)^2}{(s + 200)^2} \tag{12}$$

which is also discretized with a ZOH at 200 Hz. The closed-loop frequency response (Fig. 4b) is generated with input $z_r(t)$ (mm) as

$$z_r(t) = 0.4\sin(2\pi f t) + 1.4 \tag{13}$$

at the same set of frequencies from 0.6 to 50 Hz.

## Ball rolling algorithm and experimental setup

The ball position is captured from a USB color camera (2.9 mm Wide Angle, ELP) placed 115 cm above the surface. The image data is sampled at 260 Hz (the native camera frame rate) using a Python script and the OpenCV[61] package (Supplementary Software 1). The position of the ball is computed by finding the centroid of a color mask on the camera frame. The velocity of the ball $\dot{\mathbf{x}}_{ball}$ is derived at each time step from the calculated position and 260 Hz sample rate, and it is filtered with a five-sample moving average. The algorithm loop thus runs at 52 Hz. Given ball position $\mathbf{x}_{ball} = (i,j)$ (surface coordinates $i$ and $j$ ranging from 0 to 10) and desired goal position $\mathbf{x}_{goal}$, the positional error is

$$\mathbf{x}_{err} = \mathbf{x}_{goal} - \mathbf{x}_{ball}. \tag{14}$$

The center position of the driving shape (Fig. 5b), $\mathbf{x}_{arc}$, is

$$\mathbf{x}_{arc} = \mathbf{x}_{ball} - 0.7\hat{\mathbf{x}}_{err} + 0.2\dot{\mathbf{x}}_{ball} \tag{15}$$

where $\hat{\mathbf{x}}_{err}$ denotes the normalized vector of the positional error $\mathbf{x}_{err}$. The shape function over the surface $\mathbf{z}_r(\mathbf{x})$ (mm) is

$$\mathbf{z}_r(\mathbf{x}) = 13\exp\left(\frac{-(||\mathbf{x}_{rel}|| - 1.2)^2}{2\sigma^2}\right) \tag{16}$$

where $\sigma = 0.6$ is the standard deviation of the Gaussian profile, $||\cdot||$ is the Euclidean norm, and $\mathbf{x}_{rel} = \mathbf{x} - \mathbf{x}_{arc}$ for $\mathbf{x} = (x,y)\forall x,y \in \{0.5, 1.5, \ldots, 9.5\}$. Each cell uses $\mathbf{z}_r$ in closed-loop feedback $T_h$ (Fig. 5a). Equation(16) only holds true when inner product $\langle \mathbf{x}_{arc} - \mathbf{x}_{ball}, \mathbf{x}_{rel}\rangle > -0.4$; otherwise, $\mathbf{z}_r(\mathbf{x}) = \mathbf{0}_{[10\cdot10]}$. For multiple balls $\mathbf{z}_r(\mathbf{x})$ is independently computed for each ball position and the maximum value at each point $(i,j)$ from all sets is used to calculate the global $\mathbf{z}_r(\mathbf{x})$.

For the single ball experiment (Fig. 5c),

$$\{\mathbf{x}_{goal}\} = \{(2.5, 7.5), (7.5, 7.5), (7.5, 2.5), (2.5, 2.5)\} \tag{17}$$

The ball advances to each goal position only when $\mathbf{x}_{err} \leq 0.33$. In the second experiment (Fig. 5d), each ball is rolled down an incline at[5,10] onto the surface. The order of the balls (by color) was random. The goal positions are (0, 5) for the red ball, (0, 5) for yellow, and (5, 10) for green.

## Data availability

All relevant data from this study are included in the paper and Supplementary Information. Source data is deposited at https://doi.org/10.17617/3.9S0O4Q[62].

## Code availability

The code that supports this study is available as Supplementary Software 1. Further description of the code can be found in the Supplementary Methods.

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

## Acknowledgements

Funding for this work was provided by the National Science Foundation (CPS grant 1739452 awarded to M.E.R., C.K., J.S.H., and N.C.; DGE grant 1650115 awarded to B.K.J., V.S.) and by the Max Planck Society (C.K.).

## Author contributions

M.N. led the design of the shape display; all authors contributed towards its design and conceptualization. S.K.M., E.A., and N.K. designed the HASEL actuator. M.N and S.K.M. designed the voltage control scheme, HASEL driver circuit, and power supply system. V.S. and K.L. designed the sensor system and circuitry. M.N. wrote the firmware, software, and communications framework. B.K.J., M.N., and A.V. designed the closed loop control laws. M.N. designed the mechanical components of the shape display and integrated all subsystems. M.N., B.K.J., V.S., K.L., and A.V. built the shape display. M.N., B.K.J., V.S., K.L., and A.V. developed the experimental methodology and analyzed data; M.N., B.K.J., V.S., A.V., and K.L. wrote code for data collection and analysis; B.K.J., M.N., V.S., K.L., and A.V. implemented demos and experiments. M.N., B.K.J., and V.S. designed the figures; B.K.J. performed the photography and videography; M.N and B.K.J. created the illustrations. M.E.R., C.K., N.C., and J.S.H. supervised and administrated the project. The paper was written by B.K.J., M.N., K.L., V.S., and A.V.; all authors contributing to editing.

## Competing interests

E.A., S.K.M., N.K., and C.K. are co-founders of Artimus Robotics Inc., a company focused on commercializing HASEL actuators. E.A., S.K.M., N.K., and C.K. are inventors on US patent number 10,995,779, US Application No. 17/198,909, US Application No. 16/978,292, European Application EP18770604.9 A, and European Application EP3762618A4, which cover fundamentals and basic designs of HASEL actuators. V.S., K.L., N.C., and M.E.R. are inventors on U.S. Patent Application No. 17/746,427, "Embedded Magnetic Sensing Method for Soft Actuators". M.E.R. is a co-founder of Aspero Medical, Inc., a start-up company that is focused on commercializing balloon overtube products for use in enteroscopy. The remaining authors declare no competing interests.
