## [Peer Review File · Nature Communications]

A multifunctional soft robotic shape display with high-speed actuation, sensing, and controlREVIEWER COMMENTS

Reviewer #1 (Remarks to the Author):

Johnson and co-authors report a self-sensing reconfigurable shape display, driven by soft actuators. Each element in the 10x10 array consists of stack of 12 HASEL actuators and a magnetic sensor. The drive electronics and closed loop control are the main enabling aspect of the array. The integration is impressive. The authors show a number of polished demonstration tasks, such as moving a ball, sorting balls, or weighing fruit.

The paper is very well written, the figures are polished. The general concept will be of interest to the broad readership if Nature Communications. Performance is good for a soft system, but the benefits of compliance could be made clearer.

1. My biggest concern is the lack of detailed comparison to the 2013 inFORM project from MIT. See <https://tangible.media.mit.edu/project/inform/> and associate publications. Eg ref [3] is this manuscript

The inFORM array has at least 10x higher stroke, much higher resolution, can move heavier objects, and also operates in closed loop. It also has downsides (size). The authors should more properly place their work in context. How is soft important? What bandwidth matters? inFORM has a number of interactive demos . to what extent could this be reproduced with the 10x10 HASEL display ?

2. 60mm x 60 mm: the taxel size seems large. Why? It would be more interesting to have 10 mx10 mm. the authors claim it is possible to make smaller. So why do not do it? What is limiting factor? The HV electronics? Yield? lower stroke for smaller size? Kindly comment on what happen as you scale down.

3. Please be more balanced and also mention downsides of your methods. For example, what are downsides of magnetic sensing. Eg can you manipulated a steel ball? Or an object with an embedded magnet?

4. Scaling: what innovations would be needed for 200 x 200 display?

5. What is the average power consumption when handling a ball? This number is lacking from the paper. Put this with the claim of possible battery operation

Minor: why a 15 degree polynomial? Was 5 not enough?

Reviewer #2 (Remarks to the Author):

- What are the noteworthy results?

The primary noteworthy results of this work are the integration of sensing features for positioning and magnetic fields along with the multiplexing of many independent high voltage actuators from a single power supply operating at relatively high response frequencies. The level of integration and functionality is impressive.

- Will the work be of significance to the field and related fields? How does it compare to the established literature? If the work is not original, please provide relevant references.

The work is unique primarily due to the type of actuator used, which is a relatively recent innovation from several of the co-authors. The concept of morphing displays has been done

previously from multiple groups and is not in and of itself that unique, but the utility of the HASEL actuation is new at such a large scale. It is a substantial engineering achievement, but didn't at first glance appear to be that much improved over some previous designs. Only after reading about the technical challenges in powering and control was the novelty really apparent and most of the innovations are on the software and feedback I think that I am not completely confident in judging.

- Does the work support the conclusions and claims, or is additional evidence needed?

Yes, the work supports the conclusions. There is a probably a bit of debate over how useful such a display really could be – the performance of actual motion up and down is smaller than previous systems, the size is large and resolution not terribly high, so it serves primarily as an interesting proof of concept for fast morphing displays. I imagine if this was coupled with visual display via projection or a stretchable OLED type display rather than silicone it would be better still, but this isn't in scope for the authors effort.

- Are there any flaws in the data analysis, interpretation and conclusions? - Do these prohibit publication or require revision?

Nothing substantial that I can see, although my confidence in the electrical engineering aspects of this are lower than the mechanical components. A few minor questions stand out regarding the discussion of internal damping and its dependence on frequency.

- Is the methodology sound? Does the work meet the expected standards in your field?

Yes

- Is there enough detail provided in the methods for the work to be reproduced?

Yes, with the caveat that HASEL actuators (and high voltage actuators in general) and assembly are still somewhat challenging to complete without failures/shorts and the equipment used in this work is very specialized. Very few groups would have the expertise or systems on hand to reproduce this work quickly.

General comments:

This article shows the implementation of HASEL actuators into a 10x10 haptic display and demonstrates feedback, sensing of position/magnetic fields, vision assisted control over positioning of objects and a few other tasks that have been challenging. The article is well written, shows many examples of how their morphing display can be used, and the movies and supporting information are convincing. I have only minor changes as suggestions before publication.

The system obviously can get very complex as the number of actuators increases and the system is already quite large. What is the display resolution that could reasonably be manufactured with current HASEL technology? Is the minimum pixel size something that can be reduced to a braille type system or is the manufacturing technology still new enough that smaller resolutions are not feasible?

The oil that is used in this system is different that what I have read in the authors previous papers. Can you provide a quick overview for the readers to explain the relative advantages of each and why there is a new dielectric fluid chosen? It also wasn't clear to me from the SI documentation how much fluid was in each pouch, whether that was a variable that needed control (ie were air bubbles or non-uniform oil volumes a concern).

From the Figure 1, I didn't get a great idea of the relative dimensions of these HASEL actuators vs. the supporting circuitry and power supply. The overall height is 90 mm which is large compared to the displacement of the actuators, but the strains listed in your previous work for the actuators themselves can be around 100% in this configuration. What do you expect to be the thinnest this system could be for the same actuation displacement? Would there be any substantial challenges in moving the control circuitry to locations outside the actual display?

In the discussion of your frequency response, you describe the system becoming damped at 20-50 Hz, but don't explain why damping is only at the higher frequencies at not the lower (at least this seemed the implication of the statement). A more detailed description of what is happening here would be appreciated – I didn't follow exactly how this was the same dynamic experiment as your reference 24 and there was only a description of changing damping between the on and off part of a cycle from what I could see, rather than a frequency dependent value. Please clarify this section.

Your magnetometer was used to sense force/displacement, and you fit your experiments to a 15-degree polynomial. Is there any physical reason for the complexity of this polynomial fit? I am not sure how much might be an overfit or if this legitimately was necessary because of the physical phenomenon. Do you have to calibrate every individual actuator and if so, how much variation in the fits were there between each actuator? If many of these displays are produced, do you expect to have to calibrate every single pixel?

Is there an overall cost breakdown or bill of materials for the full display and overall estimates of power consumption? There are many advantages to using HASELs but for the size and complexity it could be perhaps valuable to provide information about this to provide guidance for future improvements and better cost effectiveness of the system.

HASELs are defined in their acronym as self-healing, but that from my understanding worked mostly with the elastomeric system that was used originally. How self-healing are these thermoplastic designs? Are there ever any leaks, or if dielectric breakdown occurs, how long do they take to recover?

Department of Mechanical Engineering
University of Colorado at Boulder
Boulder, Colorado USA

Phone: +49-1523-140-8522
E-mail: brian.k.johnson@colorado.edu

Robotic Materials
Max Planck Institute for Intelligent Systems
Stuttgart, Germany

Dear reviewers,

The manuscript "A multifunctional soft robotic shape display with high-speed actuation, sensing, and control" has been revised based on your helpful comments. Paper alterations are highlighted in **yellow** in the resubmitted proof.

Detailed below are all the specific comments addressed (*italicized*) along with our responses (**bold**).

Reviewer #1 (R1) comments:

R1C1:

“Johnson and co-authors report a self-sensing reconfigurable shape display, driven by soft actuators. Each element in the 10x10 array consists of stack of 12 HASEL actuators and a magnetic sensor. The drive electronics and closed loop control are the main enabling aspect of the array. The integration is impressive. The authors show a number of polished demonstration tasks, such as moving a ball, sorting balls, or weighing fruit.

The paper is very well written, the figures are polished. The general concept will be of interest to the broad readership if Nature Communications. Performance is good for a soft system, but the benefits of compliance could be made clearer.”

Thank you for the positive feedback and review. Regarding the benefits of compliance, we have addressed this in comment R1C2 (see below).

R1C2:

“My biggest concern is the lack of detailed comparison to the 2013 inFORM project from MIT. See <https://tangible.media.mit.edu/project/inform/> and associate publications. Eg ref [3] is this manuscript The inFORM array has at least 10x higher stroke, much higher resolution, can move heavier objects, and also operates in closed loop. It also has downsides (size). The authors should more properly place their work in context. How is soft important? What bandwidth matters? inFORM has a number of interactive demos . to what extent could this be reproduced with the 10x10 HASEL display ?”

Thank you for this question. While it is important to place our work in context, we also do not seek to compete specifically with inFORM as it is only one of many types of shape display cited. We have revised some of the Results and Discussion to add further context and address some of these questions. In addition to being smaller and more portable, the soft display has a higher maximum strain in comparison to total device height (12 mm/90 mm = 13.3% after 60 cycles, inFORM 100 mm/1100 mm = 9.1% [r1]). inFORM reports a maximum force of 1.08 N per pixel (regardless of pixel size), while each HASEL actuator can exert roughly 2.5 N (varies with pixel size). Power consumption is similar per actuator compared to inFORM (see also R1C6) [r1]. The magnetic sensing of the soft surface enables force sensing and new interactivity not possible with inFORM (e.g. Fig 4). Softness allows for surface continuity and new tactile experiences which differ from inFORM’s pixelated surface with rigid edges and corners. It also enhances the distribution of forces across the surface for object sensing applications (e.g. Fig 3). With a higher bandwidth, sTISSUE

can create vibrational tactile feedback (requires minimum 40 Hz vibration [r2]), which can be a useful bandwidth metric. While some inFORM demos cannot be replicated with the display (e.g. pulling a pixel pin), many inFORM demos rely on a Microsoft Kinect camera to incorporate user position feedback; similar interactive demos could be reproduced on the soft surface if a Kinect was added, but it is outside the scope of our work.

[r1] Follmer, S., Leithinger, D., Olwal, A., Hogge, A. & Ishii, H. inFORM: dynamics physical affordances and constraints through shape and object actuation. *in Proc. UIST* (2013).

[r2] Chouvardas, V. G., Miliou, A. N. & Hatalis, M. K. Tactile displays: overview and recent advances. *Displays* 29 (2008).

R1C3:

“60mm x 60 mm: the taxel size seems large. Why? It would be more interesting to have 10 mx10 mm. the authors claim it is possible to make smaller. So why do not do it? What is limiting factor? The HV electronics? Yield? lower stroke for smaller size? Kindly comment on what happens as you scale down.”

Thank you for the questions. We would like to clarify that the actuator size is roughly 58 mm x 49 mm, one side longer due to a longer skirt of the actuator film. There is a 11 mm gap (including the film skirt on the long side) between actuators which results in the total 60 mm x 60 mm footprint. The choice of taxel size represents a tradeoff between manufacturability, scalability, and utility; the larger size enables the surface to interact with objects at a macro scale (like human interaction, rolling a ball, or shaking a beaker of liquid (Fig. 1)) using only a 10x10 array. A 10 mm x 10 mm size in a 10x10 array is a small area, and increasing the scale of the display (e.g. 200 x 200, see R1C5) is a technical challenge more than a scientific challenge. However, we show that even a 10x10 array provides sufficient shape control authority (like accurately moving a ball or displaying text). On the other hand, fabricating HASELs at a size of 10 mm x 10 mm becomes very difficult and prone to error since today’s fabrication methods are mainly done by hand. A smaller actuator produces smaller stroke and is limited by the stiffness of the actuator film [r3], but this could be offset by increasing the number of pouches per actuator. Another challenge is that the magnetic block for the sensing mechanism becomes less sensitive with smaller size, resulting in larger sensor noise and less sensing resolution. However, the electronics themselves do not pose a limitation on smaller size; the height of the overall device must simply increase to fit the electronics into the smaller footprint. The paper has been revised with these details in the Results and Discussion.

[r3] Rothmund, P., Kellaris, N., Mitchell, S. K., Acome, E. & Keplinger, C. HASEL artificial muscles for a new generation of lifelike robots—recent progress and future opportunities. *Adv. Mater.* 33, 2003375 (2021).

R1C4:

“Please be more balanced and also mention downsides of your methods. For example, what are downsides of magnetic sensing. Eg can you manipulated a steel ball? Or an object with an embedded magnet?”

We appreciate this feedback and to address this have revised the paper in the Results and Discussion sections. Soft magnetic materials including steel will have an influence on the magnetic field and thus sensing. On one hand, this allows the distributed sensing array to measure the position of the object (similar to our demonstration in Fig 4 which detected the position of the embedded magnet in the wand device), which could be used for object manipulation. Conversely, the disturbance of magnetic field makes the measurement unsuitable for displacement feedback control; only the voltage regulation can be used to generate open loop displacements of each actuator. If the soft surface is moved (e.g. across the room), the sensors will need to be recalibrated due to the change of the local ambient magnetic field; however, the recalibration time is less than 10 minutes and does not increase with array size (e.g. 200 x 200) since multiple actuators can be

calibrated simultaneously.

R1C5:

“Scaling: what innovations would be needed for 200 x 200 display”

We appreciate this feedback and have added further details in the Results and Discussion to clarify scaling requirements. Significant scientific innovations are not required to scale to a 200 x 200 display; technical implementation is the main challenge. Each 1x10 module can be scaled to an indefinite amount with two caveats: first, additional power sources beyond the single AC/DC power supply will be required as the number of modules increase; secondly, a new communication and control structure must be implemented. Currently, the PC sends and receives data from each 1x10 module via USB hubs and performs multithreaded computation to maintain a 600 Hz communication rate. Scaling beyond a 10x10 display will require either a computer processor with many additional cores to maintain 600 Hz speed or multiple computers which coordinate control together through dozens of USB hubs. Additional software would be required to handle the large amount of USB inputs and data.

R1C6:

“What is the average power consumption when handling a ball? This number is lacking from the paper. Put this with the claim of possible battery operation”

Thank you for highlighting this deficiency. We have revised the Results section with addition power consumption information. The average power consumption itself is difficult to measure without impacting the system performance. To provide an estimate, each high voltage (HV) power supply which powers ten HASELs outputs a maximum of 24 W (3mA at 8 kV), and each HV driver circuit consumes about 0.4 W at peak load (max charge or discharge rate). This corresponds to an average consumption of 2.8 W per actuator at peak load (high-frequency, high-amplitude actuation). Handling a ball requires only a small portion of actuators at any given time (about 9-12 spread across three 1x10 modules), so the estimated consumption would be less than $3*24+0.4*12 = 76.8$ W at peak load. Peak load only occurs while increasing actuator stroke; maintaining or decreasing stroke uses little power, so the average consumption will be significantly lower than 76.8 W. The HV power supply itself draws a maximum of 1.6 A at 24 V, well within the range of commercially available lithium polymer batteries (a 16000 mAh battery can power all 100 actuators at continuous peak load for one hour).

R1C7:

“Minor: why a 15 degree polynomial? Was 5 not enough?”

Thank you for the question. We believe a 5-degree polynomial would be sufficient, but we include the full polynomial which was implemented in the software.

Reviewer #2 (R2) comments:

R2C1:

“This article shows the implementation of HASEL actuators into a 10x10 haptic display and demonstrates feedback, sensing of position/magnetic fields, vision assisted control over positioning of objects and a few other tasks that have been challenging. The article is well written, shows many examples of how their morphing display can be used, and the movies and supporting information are convincing. I have only minor changes as suggestions before publication.”

We appreciate this positive feedback in addition to Reviewer #2's thorough review of the paper (not reproducing here for length).

R2C2:

“The system obviously can get very complex as the number of actuators increases and the system is already quite large. What is the display resolution that could reasonably be manufactured with current HASEL technology? Is the minimum pixel size something that can be reduced to a braille type system or is the manufacturing technology still new enough that smaller resolutions are not feasible?”

Thank you for these questions.

- (1) The HASEL technology does not limit the resolution (scale) of the display; rather, the display resolution is a tradeoff of power and communication requirements which scale with size; we describe this further in our response for R1C5 and have revised the paper in the Discussion section to add these details.**
- (2) The minimum pixel size cannot currently be reduced to a braille-type system, as research shows that at mm-scale sizes, the effectiveness of HASEL actuators become limited by the thickness and stiffness of the films used in their construction [r3]. As discussed in R1C3, fabrication of small actuators is also challenging due to several precise steps which are not yet automated today.**

[r3] Rothmund, P., Kellaris, N., Mitchell, S. K., Acome, E. & Keplinger, C. HASEL artificial muscles for a new generation of lifelike robots—recent progress and future opportunities. *Adv. Mater.* 33, 2003375 (2021).

R2C3:

“The oil that is used in this system is different that what I have read in the authors previous papers. Can you provide a quick overview for the readers to explain the relative advantages of each and why there is a new dielectric fluid chosen? It also wasn’t clear to me from the SI documentation how much fluid was in each pouch, whether that was a variable that needed control (ie were air bubbles or non-uniform oil volumes a concern).”

We appreciate this feedback and have added an explanation on the choice of dielectric fluid and the volume of fluid used in the Supplementary Information. To summarize here, we used a silicone liquid dielectric which has a lower viscosity than the dielectrics used in previous works, giving better dynamic performance. Each HASEL pouch was filled with 0.4 mL dielectric fluid, and each actuator contains 24 pouches, for a total of 9.6 mL fluid. Large air bubbles were removed, but small (below 3 mm diameter) bubbles are not a concern. Non-uniform oil volumes (from 0.35-0.45 mL) are also not a concern because this has only a small impact on actuator displacement.

R2C4:

“From the Figure 1, I didn’t get a great idea of the relative dimensions of these HASEL actuators vs. the supporting circuitry and power supply. The overall height is 90 mm which is large compared to the displacement of the actuators, but the strains listed in your previous work for the actuators themselves can be around 100% in this configuration. What do you expect to be the thinnest this system could be for the same actuation displacement? Would there be any substantial challenges in moving the control circuitry to locations outside the actual display?”

Thank you for the question. We have amended the Results and Fig 1 to provide more detail on dimensions. The height of each unpowered actuator is roughly 14 mm, resulting in an actuator strain of approximately 86% after 60 cycles. One factor that reduces strain is the elastic force of the surface skin, which acts as a spring load on the actuator. We do not anticipate any substantial challenge in moving the circuitry outside of the display, which could potentially reduce the system thickness to 30-40 mm overall. The downside of this approach is that the device becomes less compact (in terms of overall footprint) and less portable as significant external wiring is required.

R2C5:

“In the discussion of your frequency response, you describe the system becoming damped at 20-50 Hz, but don’t explain why damping is only at the higher frequencies at not the lower (at least this seemed the implication of the statement). A more detailed description of what is happening here would be appreciated – I didn’t follow exactly how this was the same dynamic experiment as your reference 24 and there was only a description of changing damping between the on and off part of a cycle from what I could see, rather than a frequency dependent value. Please clarify this section.”

We appreciate the feedback. We have added a revision to the Results to clarify why the system becomes damped. The reason is primarily due to the inertial and viscous mechanics of actuation, although inertial mechanics dominate for this actuator geometry [r4]. Positive deformations are induced by fluid being displaced from beneath the electrode area, and negative deformations induced by fluid flowing back to the region. The fluid is limited in the speed at which it can flow between these two modes: under high frequency voltage inputs (>20 Hz) this causes a reduction in overall actuator deformation amplitude.

[r4] Rothemund, P., Kirkman, S. & Keplinger, C., Dynamics of electrohydraulic soft actuators. Proceedings of the National Academy of Sciences 117, 16207–16213 (2020).

R2C6:

“Your magnetometer was used to sense force/displacement, and you fit your experiments to a 15-degree polynomial. Is there any physical reason for the complexity of this polynomial fit? I am not sure how much might be an overfit or if this legitimately was necessary because of the physical phenomenon. Do you have to calibrate every individual actuator and if so, how much variation in the fits were there between each actuator? If many of these displays are produced, do you expect to have to calibrate every single pixel?”

Thank you for the question. We have addressed this in R1C7, however, we also wish to clarify that the displacement fit is a 3-degree polynomial (coefficients unique to each actuator), while the force-sensing fit is the 15-degree polynomial (the same for all actuators). There is sufficient variation between fits that it is necessary to calibrate each actuator individually for the displacement mapping. Calibration takes approximately 8 seconds per pixel, but multiple actuators can be calibrated simultaneously. The calibration remains accurate for many weeks of operation; recalibration is only necessary if the display is moved. We have revised the Results and Methods to clarify some of these details.

R2C7:

“Is there an overall cost breakdown or bill of materials for the full display and overall estimates of power consumption? There are many advantages to using HASELs but for the size and complexity it could be perhaps valuable to provide information about this to provide guidance for future improvements and better cost effectiveness of the system.”

Thank you for asking about these details. We have addressed questions of power consumption in R1C6. Regarding cost and materials breakdown, we have revised the Supplementary Information to provided further details.

R2C8:

“HASELs are defined in their acronym as self-healing, but that from my understanding worked mostly with the elastomeric system that was used originally. How self-healing are these thermoplastic designs? Are there ever any leaks, or if dielectric breakdown occurs, how long do they take to recover?”

We appreciate the question. The thermoplastic film is not self-healing, but breakdown within the dielectric fluid is self-healing. If breakdown occurs, it can cause pinhole leaks in the film, which

typically affect only one pouch at a time (slight performance decrease); only multiple leaks will require the actuator to be replaced. Often, breakdown does not cause any permanent damage and there is no recovery period; operation can be resumed immediately. Due to this, we describe the design as self-healing and wish to stay consistent with previous literature.

REVIEWERS' COMMENTS

Reviewer #1 (Remarks to the Author):

The referee comments have been satisfactorily addressed.

Reviewer #2 (Remarks to the Author):

All comments have been addressed adequately and the paper can be published as is. Good work and I'll look forward to seeing future developments.

Department of Mechanical Engineering
University of Colorado at Boulder
Boulder, Colorado USA

Phone: +49-1523-140-8522
E-mail: brian.k.johnson@colorado.edu

Robotic Materials
Max Planck Institute for Intelligent Systems
Stuttgart, Germany

Dear reviewers,

The manuscript "A multifunctional soft robotic shape display with high-speed actuation, sensing, and control" has been revised for publication. Paper alterations are appear using Microsoft Word Track Changes in the resubmitted proof.

Detailed below are all the specific comments addressed (*italicized*) along with our responses (**bold**).

Reviewer #1 (R1) comments:

R1C1:

"The referee comments have been satisfactorily addressed."

Thank you for your previous feedback; we are happy to have satisfactorily addressed your comments.

Reviewer #2 (R2) comments:

R2C1:

"All comments have been addressed adequately and the paper can be published as is. Good work and I'll look forward to seeing future developments."

We thank you for your previous feedback and thank you for the positive comments.